# Microparticles of Sericin-Dextran Conjugate for Improving the Solubility of Antiviral Drug

**DOI:** 10.3390/jfb14060292

**Published:** 2023-05-24

**Authors:** Shuqi Chen, Xiaolong Feng, Xinwei Li, Miaochang Liu, Wenxia Gao, Qian Miao, Huayue Wu

**Affiliations:** College of Chemistry and Materials Engineering, Wenzhou University, Wenzhou 325027, China

**Keywords:** sericin, sericin-dextran conjugate, microparticles, atazanavir, solubility

## Abstract

A novel sericin-dextran conjugate (SDC) and self-assembled microparticles has been prepared for improving solubility of atazanavir. Microparticles of SDC were assembled by the reprecipitation method. The size and morphology of SDC microparticles could be adjusted by the concentration and solvents. Low concentration was conducive to the preparation of microspheres. Heterogeneous microspheres could be prepared in ethanol with the range of 85–390 nm, and hollow mesoporous microspheres in propanol with an average particle size of 2.5–22 µm. The aqueous solubility of atazanavir was improved to 2.22 mg/mL in buffer solutions at pH 2.0 and 1.65 mg/mL at pH 7.4 by SDC microspheres. In vitro release of atazanavir from hollow microspheres of SDC exhibited a slower release, had the lowest linear cumulative release in basic buffer (pH 8.0), and the most rapid double exponential diphase kinetic cumulative release in acid buffer (pH 2.0).

## 1. Introduction

Virus infection has threatened seriously health and quality of life, such as HIV, hepatitis virus, influenza virus, and the global pandemic of COVID-19 pneumonia and the outbreak of monkeypox have been serious challenges to the global public health. The development of new efficient antiviral drugs was significant. Although the number of newly developed antiviral active chemicals has increased with advances in chemical synthesis technologies, mostly synthetic antiviral drugs were insoluble in water, such as ritonavir, lopinavir, tipranavir, acyclovir, atazanavir, amprenavir [1]. For example, amprenavir, a non-peptide protease inhibitor of taurine, has a aqueous solubility of only 0.04 mg/mL. The poor solubility of antiviral drug affected their bioavailability [2]. Poor water solubility of many active ingredients was one of the major challenges in drug development, with approximately 40% of approved drugs and nearly 90% of lead compounds having poor aqueous solubility [3]. Many methods have been tried to overcome limited aqueous solubility, such as nanocrystalline technology, inclusion technology, polymer-based amorphous solid dispersions technology, polymer micelle and so on [4]. Polymer micelles were self-assembled from amphiphilic block copolymers, which could encapsulate and load insoluble drugs. Polymeric micelles have many advantages, such as good dispersion, high drug loading, long cycle time high permeability and retention effect, and so on [5]. Polymer-protein conjugates were a class of amphiphilic polymeric biohybrids with unique properties [6]. They were important in biomedicine ranging, not only for improving the stability, elongating the circulation time and lowering the immunogenicity of protein drugs, but also for drug carrier, nano drug delivery systems, biomedical imaging, and so on [7,8,9]. In this study, a novel sericin-dextran conjugate (SDC) and its self-assembled microparticles have been prepared for improving solubility of antiviral drugs in this study.

Sericin is one of the two important components of silk, which accounts for 20–30% of the total silk protein [10]. Sericin is widely used in cosmetics because it is rich in polar amino acids (serine, glutamic acid, tyrosine, aspartic acid, etc.) and it has obvious water solubility and high reactivity, and also has the characteristics of anti-oxidation and anti-ultraviolet [11]. Sericin also has great potential in pharmaceutical and biomedical applications, such as drug delivery systems [12].

However, sericin is often discarded with waste water in the process of silk gumption due to its poor mechanical support and poor formability, it hinders the use of sericin as a recycled material [13]. Moreover, silk proteins are limited as the drug releasing matrix, because the rate of drug release from it is difficult to control. Antiviral and chemotherapy drugs are mostly insoluble drugs [14]. A technique of combining sericin chitosan, hyaluronic acid and synthetic polymer polyvinyl alcohol (PVA) into block copolymer micelles can simultaneously solve the application problems of sericin and insoluble drugs, and effectively improve the solubility and bioavailability of drugs [15,16,17].

Dextran was chosen as polymer backbone in this paper to prepare sericin-dextran conjugate (SDC) as drug carrier, since dextran has been widely used as bioconjugation with bioactive molecules, such as enzymes, proteins, drugs, and other bioactive molecules to improve bioavailability and physicochemical properties [18].

SDC was synthesized and then self-assembled into microparticles. These microparticles of SDC were investigated for in vitro release of atazanavir, a HIV protease inhibitor. Atazanavir has low bioavailability due to the poor water solubility and rapid first-pass metabolism in liver [19]. The results showed that these SDC microparticles helped to improve solubility of atazanavir and also realized controlled release.

## 2. Experimental Part

### 2.1. Materials

All solvents and reagents were analytical reagents without purification. Dextran-5 was obtained from Shanghai Huamao Pharmaceutical Co., Ltd. (Shanghai, China). Sick sericin was purchased from Huzhou Attest Biochemical Co., Ltd. (Huzhou, China). Molecular weights of dextran and silk sericin were determined by GPC with Waters-e2695. *N,N′*-carbonyldiimidazole (CDI) was obtained from Sinopharm Chemical Reagent Co. Ltd. (Shanghai, China).

### 2.2. Measurements

^1^H NMR spectra was acquired using a 300-Bruker spectrometer (300 MHz) (Bruker, Mannheim, Germany) and reported in parts per million (ppm) relative to the internal standard TMS. FT-IR spectra was recorded on a Bruker EQUINOX 55 spectrometer with the KBr-technique. Molecular weight was determined by gel permeation chromatography (GPC) with Waters-e2695 (Waters, Milford, MA, USA).The viscosity of dextran, sericin, and dextran-sericin conjugate solutions in water (5.0% *w*/*w*) was determined with a rotation viscosimeter (NDJ-8S, Shanghai Changi, Shanghai, China). Differential scanning calorimeter (DSC) and thermo-gravimetric analysis (TGA) was performed on a simultaneous TGA/DSC instrument (SDT Q600, TA Instruments, New Castle, DE, USA), at a heating rate of 10 °C min^−1^. Powder X-ray diffraction (XRD) measurements were carried out on a Bruker D8 Advance diffractometer (Bruker, Germany). The samples with Cu Kα radiation (λ = 1.5418 A) were detected at a voltage of 40 kV and a current of 200 mA. Scanning electronic microscopy (SEM) images were carried out by a Nova NanoSEM 200 scanning electron microscope (FEI, South Moravian Region, Czech Republic). The size and distribution of the microparticles were performed by dynamic light scattering (DLS) (Zetasizer Nano-ZS, Malvern, UK) at 25 °C. High-performance liquid chromatography (HPLC, Agilent 1120 Compact LC system (Agilent, Santa Clara, CA, USA)) was performed with Inertsil ODS-SP C18column (4.6 mm × 250 mm, 5 µm).

### 2.3. Preparation of Sericin-Dextran Conjugate

Sericin-dextran conjugates were synthesized as Figure 1. Sericin (0.60 g) and 0.62 g dextran were added to DMSO (25 mL), then 0.62 g CDI was added to react at 80 °C for 12 h with stirring under nitrogen protection. The mixture was precipitated in isopropanol. The crude product was washed twice with ethanol to get sericin-dextran conjugate.

### 2.4. Microparticles Preparation

Sericin-dextran conjugate microparticles were prepared by the reprecipitation method. The compound was added in deionized water, and the solution (0.1 mL) was injected into vigorously stirred organic solvents (4 mL) using a micro-syringe, at a given temperature. SDC microparticles were prepared in different organic solvents and with different concentrations of SDC (5%, 10%, 15%, and 20%). The microparticles were centrifugated and lyophilized for 8 h. They were sputtered with gold for SEM images.

### 2.5. In Vitro Release

Drug loaded SDC microparticles were prepared using evaporation and mechanical abrading. 10 mg atazanavir was dissolved in 1.0 mL anhydrous ethanol, 100 mg SDC microparticles was added. The solid dispersant of atazanavir-loaded SDC microparticles was obtained by grinding the mixture until dryness. The tablet for the drug-loaded microparticles and native drug was prepared with a 7 mm die, using a compression force of 10 kN. It was then incubated in buffer solution (pH = 2.0, 6.5, 7.4, 8.0) at 37 ± 0.5 °C with the stirring speed of 100 rpm. 0.5 mL solution was extracted at selected intervals and filtered with a 0.22 µm membrane for HPLC detection. The quantitative analysis of atazanavir using HPLC was referred to the reported method [20]. The mobile phase was comprised of 0.2% ammonium dihydrogen phosphate aqueous solution, and the mixture of phosphoric acid and acetonitrile (1:1), it was adjusted to pH 2.5. The flow rate was set to 1.5 mL/min, and the sample size was 20 µL. The wavelength for UV detection was 288 nm.

### 2.6. Solubility Study

The solubility test was carried. Excess amounts of each sample were placed in buffer solution (pH = 2.0, 6.5, 7.4, 8.0) (50 mL) at 37 °C under stirring in an orbital incubator (60 rpm). 2 mL samples were withdrawn and replaced with equal volumes of fresh fluid. The samples were centrifuged and tested by HPLC in triplicate.

## 3. Results and Discussion

### 3.1. Preparation of Sericin-Dextran Conjugate

The infrared spectra of dextran, sericin, and sericin-dextran conjugate were presented in Figure 1. Figure 1a showed that dextran had six characteristic peaks at 3420, 2925, 1647, 1047 and 546 cm^−1^. Among them, 3420 cm^−1^ was attributed to the stretching vibration of a large amount of hydroxyl group (-OH), 2925 cm^−1^ to the -CH_2_- symmetric stretching vibration, 1647 cm^−1^ to the C-O stretching vibration, 1047 cm^−1^ to the α-(1→6) glycosidic bond, and 546 cm^−1^ to the pyranose ring skeleton [21]. Figure 1b showed that the characteristic peaks of sericin are 3320 cm^−1^ and 3067 cm^−1^ (N-H), 1659 cm^−1^ (amide I), 1533 cm^−1^ (amide II), 1251 cm^−1^ (amide III) and 648 cm^−1^ (amide V) [22]. From the spectrum of sericin/glucan mixture (Figure 1c), it could be seen that the stretching vibration peak of a large number of hydroxyl (-OH) groups in dextran, the absorption peak of α-(1→6) glucoside bond at 1047 cm^−1^, and the stretching vibration peak of pyranose ring skeleton at 546 cm^−1^. There were characteristic peaks of 1659 cm^−1^ (amide I), 1533 cm^−1^ (amide II) and 1251 cm^−1^ (amide III) in sericin. The above results were similar to the native polymer spectra of dextran and sericin. The sericin-dextran conjugate showed absorption bands at 2929, 1659, 1535, 1402, 1263, 1151, 1014, 761 and 552 cm^−1^ (Figure 1d), new bands appeared at 1014 cm^−1^ and 1151 cm^−1^ compared with sericin, belonging to the C-O-C stretching vibration in the -COO, confirmed the successful synthesis of the sericin-dextran conjugate.

Figure 2 was the ^1^H NMR spectra of dextran, sericin, and sericin-dextran conjugate. Figure 2a showed the ^1^H NMR spectrum of dextran, the proton peaks at 4.99 ppm belonged to the 1-H proton characteristic peak, 4.00 ppm belonged to the 6-H, 3.91 ppm belonged to the 5-H, 3.70–3.77 ppm belonged to 3, 4-H proton peak, and 3.52–3.59 ppm belonged to 2-H proton characteristic peak. Figure 2b showed the ^1^H NMR spectrum of sericin, the proton peaks at 6.82, 6.88, 7.19 and 7.28 ppm belonged to the aromatic ring signal in the tyrosine residue, 8.44 and 8.55 ppm belonged to the proton characteristic peak of-(CONH) in the tyrosine and serine in sericin. The proton peaks at 3.89, 4.01 and 4.52 ppm belonged to the characteristic peak of -CH in tyrosine and serine. The proton peaks at 2.07, 2.78 and 2.99 ppm belonged to the characteristic peak of -CH_2_ in serine and aspartic acid. Figure 2c showed the ^1^H NMR spectrum of the sericin-dextran conjugate. The proton peaks belonging to the aromatic ring of tyrosine residues in sericin shifted from 6.82, 6.88, 7.19, 7.28 ppm to 6.83, 7.12, 7.29, 7.47 ppm. Moreover, the characteristic peak of -(CONH) in tyrosine residues also decreased from 8.44 ppm to 8.61 ppm, and from 8.55 ppm to 8.69 ppm. The shift of these proton peaks might be the result of the shielding effect of the triazine ring on the tyrosine residue [23]. The 1-H proton peak at 4.99 ppm and the 2, 3, 4-H proton peak at 3.51–3.76 ppm of dextran were also observed in the spectrum. These results indicated the successful synthesis of sericin-dextran conjugate.

The structure of sericin-dextran conjugate was further characterized by XRD spectra, as shown in Figure 3. Dextran has a section semi-crystalline structure, which shows a weak diffraction peak at approximately 20.6° and 22° of 2θ. The crystallinity of dextran was mainly attributed to the formation of intermolecular and intramolecular hydrogen bonds, which are easily affected by the introduction of functional groups [24]. The sericin showed a characteristic peak at approximately 17.7°, the secondary structure of sericin was mainly random curling and less β-conformation, with microcrystalline [25].When the hydroxyl of dextran participated in the graft reaction, the hydroxyl of dextran reacted with the functional groups such as hydroxyl and carboxyl of serine (Ser) and aspartic acid (Asp) in sericin. Then the content of hydroxyl both of amorphous and crystalline regions gradually decreased with the participation in the modification reaction. So, the characteristic peaks of dextran and sericin were migrated to 19.4° for the sericin-dextran conjugate. These changes suggested that the semi-crystalline dextran and sericin were destroyed and the diffraction peaks were changed during modification.

Molecular weight of sericin-dextran conjugate was further determined by GPC, the graphs of GPC were available in Appendix A. The GPC result of the polymer showed that the number-average molecular weight (Mn) and the weight-average molecular weight (Mw) of the polymer are 9814 and 13,833, and the polydispersity index was 1.4. Compared with native dextran (Mn = 5256, Mw = 8735, Polydispersity 1.66) and sick sericin (Mn = 3440, Mw = 3967, Polydispersity 1.15), the degree of polymerization of conjugate was increased. The graft ratio of sericin conjugated to dextran was close to one, according to the Mn and Mw. This sericin-dextran conjugate was further confirmed by detecting the viscosity of conjugate. The viscosity of dextran solution (5% *w*/*w*) was 4.1 mPa·s, and the viscosity of sericin solution (5% *w*/*w*) was 3.3 mPa·s. The viscosity of sericin-dextran conjugate solution (5% *w*/*w*) was increased to 9.3 mPa·s.

The DSC/TGAcurvesofdextran, sericin, and sericin-dextran conjugate were presented in Figure 4. Dextran showed endothermic peaks around 60 °C corresponding to the evaporation of bound water, and the peak around 320 °C due to the decomposition of sample (Figure 4a). The TGA indicated that dextran started to degrade around 290–350 °C, with about 70% mass loss, it was in agreement with the DSC data. As for sericin, two obvious endothermic peaks were exhibited around 70 °C and 220 °C, due to the evaporation of bound water and the decomposition of sericin, respectively (Figure 4b). The sericin-dextran conjugate started to degrade at 150 °C lower than native dextran (290 °C) (Figure 4c). This decrease was probably caused by the decrease of crystallinity dextran after conjugating with sericin [26].

### 3.2. Effect of Concentration on Microspheres’ Forming

SDC microparticles were prepared with different concentration of SDC solutions (5%, 10%, 15%, 20%). The morphology and size of SDC microparticles was affected by the concentration, as shown in Figure 5. At relatively low concentration (5% and 10%), the SDC microparticles appeared sphere-like shape, as shown in Figure 5a,b. Microparticles with the size distribution of 61–292 nm were prepared at the concentration 5%. The dynamic light scattering (DLS) revealed that the Z-average particle size was 146 nm. Microparticles prepared at the concentration 10% had the size of 85–390 nm and the Z-average particle size of 256 nm. The zeta potential measurement showed the surface potential of all microparticles lied in between −14.2 mV to −16.6 mV. There was no significant change in zeta potential after drug loading. There was a tendency to obtain greater microspheres with increasing the concentration. At relatively high solution concentration (15% and 20%), irregular micro-block were appeared (Figure 5c,d). When the concentration was too large, the molecules had no time enough to self-assemble into supramolecular structures. A large number of SDC molecules were exposed to ethanol to precipitate and then to aggregate into block immediately. Therefore, microspheres and partial micro-blocks were aggregated at the solution concentration of 15%, and totally blocky aggregations were obtained with further increase the concentration to 20%.

### 3.3. Effect of Solvents on Microparticles

In addition to the concentration affecting the formation of microspheres, the type of organic solvent, dielectric constant and viscosity also have great influence on the size and morphology of microparticles. Kinds of solvents with different dielectric constantand viscosity were studied, including methanol (*η* = 0.60 cp, *ε* = 33.0), DMF (*η* = 0.92 cp, *ε* = 38.3), ethanol (*η* = 1.20 cp, *ε* = 24.3), *n*-propanol (*η* = 2.27 cp, *ε* = 22.2),and *i*-propanol (*η* = 2.37 cp, *ε =* 18.3). SEM images of microparticles prepared by injection of SDC solution (10%) into different organic solvents were showed in Figure 6.

Miroparticles with different shape were obtained in organic solvents. Hollow microspheres (2.5–22) µm in average diameter with smooth surface outside were obtained in *n*-propanol at a high formation yield (Figure 6a). A typical sectional view of hollow microsphere was shown in Figure 6e. The hollow microsphere had the shell thickness of 1.7 µm, and had a large number of meso-macropores (40–300 nm) in both the shell wall and the housing inner wall. The outer wall of the shell of the hollow micro-ball was smooth with no porosity. So the unique structure of these hollow microspheres with rich meso-macropores inside and therefore very large specific surface area inside, would be good adsorption and storage materials. The hollow microspheres appeared varied morphology, such as the hollow microspheres with a cavity (Figure 6b), with a single concave surface (Figure 6c), with a plurality of surface depressions (Figure 6d). The hollow microspheres containing a single concave universally existed. The hollow microspheres with relatively large diameter usually had several surface depressions. The invagination produced under the pressure of the solvent outside of the hollow micro-balls, since these hollow microspheres had large hollow spherical volume, large internal cavity and rich meso-macropores inside the wall. When the pressure of the solvents was too large on the outside, a cavity was caused on the shells of hollow microspheres, or the entire hollow microspheres and their shells broke into multiple fragments.

Hollow microspheres were self-assembled in *n*-propanol with hydrophobic core and hydrophilic shell. Sericin was composed of eighteen amino acids, of which serine was the most, accounting for about one-third of the total amino acids. Sericin contained a large number of polar functional groups such as hydroxyl, carboxyl and amino groups [27]. Sericin also had non-polar amino acid residues, such as tryptophan, phenylalanine and tyrosine [28]. The amphiphilic character of these polymers guarantied the self-assembling of regular microspheres [29]. However, the self-assembling capacity of the particle depended on the balance of different polar functional groups, as well as the balance between molecular and solvent polarity [30]. Therefore, only hollow microspheres could be obtained in other solvents.

Microspheres with smooth surface were got both in ethanol (Figure 6g) and isopropanol (Figure 6h). Heterogeneous microspheres with the diameter of 85–390 nm were prepared in ethanol, and with the diameter of 105–820 nm in isopropanol. The viscosity of isopropanol is 2.37 (cp), much greater than that of ethanol. Since the penetration of SDC solution into organic solvent was slowed down with the increase of solvent’s viscosity, the process of crystallization and self-assembly process was prolonged. Therefore, the microparticles’ size increased [31]. However, the molecules did not have enough time to self-assemble into supramolecular structures in methanol and DMF with too low viscosity. In addition, the hydrogen bond interaction between SDC and organic solvent also affected the self-assembly process. When SDC solution penetrated into the strongly polar solvents such as methanol and DMF, the intervention of solvent molecules would destroy the hydrogen bond between water molecules and SDC, with the result of increasing molecular force of SDC to gather and precipitate into micron blocks [32]. Therefore, the size and morphology of the microparticles could be controlled by the solvents.

### 3.4. Solubility Study

Atazanavir (C_38_H_52_N_6_O_7_, Mw = 704.86) is an azapeptide HIV-1 protease inhibitor for HIV-1 infected patients [33]. Atazanavir is also significant in treating the COVID-19 infection for inhibiting the protease activity of COVID-19 which results in inhibition of viral replication [5]. Atazanavir has solubility of 2.16 mg/mL in acidic environments (pH 1.0), and 0.033 mg/mL (pH 3.0). It has highly pH dependent aqueous solubility. This resulted in low oral bioavailability and short half-life [20]. To improve the aqueous solubility, salt formation, inclusion complexes, and solid dispersion has been tried by reducing size or destroying the crystalline of drug. The particle size and specific surface area of drug directly affected the pharmacokinetics and biological distribution of drug molecules, which were key characteristics of nanotechnology drug delivery systems. In this research, SDC microparticles were selected as new drug carrier for improving solubility of atazanavir.

Dissolution experiments of pure atazanavir and atazanavir-loaded SDC microparticles were carried out in buffer solutions at pH 2.0, 6.5, 7.4, and 8.0, as showed in Figure 7. The release of atazanavir from the native drug tablets was a pH dependent aqueous solubility. It showed the solubility of 2.21 mg/mL at pH 1.0, and it decreased rapidly to 0.29 mg/mL with increasing pH at pH 2.0, and 0.041 mg/mL at pH 3, and it showed nearly no release with increasing pH (Figure 7A). The solubility of atazanavir was improved to 2.22 mg/mL within 194 min in buffer solutions at pH 2.0 by SDC microparticles, to 1.65 mg/mL at pH 7.4, to 1.24 mg/mL at pH 8.0 (Figure 7B). The enhanced aqueous solubility of atazanavir was possibly due to the large surface area and hydrophilic dextran and sericin. Therefore, the SDC microparticles had a great effect on the solubility of atazanavir. The solubility of atazanavir exhibited the linear release (Figure 6B), linear regression equation Y_a_ = −0.70229 + 0.01491 × t (r = 0.99654, *p* < 0.0001, Y: cumulative release), Y_b_ = −0.65499 + 0.01420 × t (r = 0.99722, *p* < 0.0001), Y_c_ = −0.40382 + 0.01066 × t (r = 0.99901, *p* < 0.0001), Y_d_ = −0.58325 + 0.01064 × t (r = 0.99248, *p* < 0.0001).

### 3.5. In Vitro Release

In vitro release studies of atazanavir based on hollow microsphere with maso -macroporosity was showed in Figure 8. A total of 51.2% of atazanavir was released at pH 8.0, 99.2% at pH 7.4, 97.8% at pH 6.5, and 94.8% at pH 2.0 after 120 h. The lowest cumulative release of atazanavir from the microparticles was investigated in basic buffer (pH 8.0), which exhibited a linear release equation of Y_d_ = −0.4186 + 0.4369 × t (R^2^ = 0.99082, Y: cumulative release). The most rapid cumulative release of atazanavir from the microparticles was investigated in acid buffer (pH 2.0), which fitted the double exponential diphase kinetics equation of Y_a_ = 99.85941 − 70.01088 × e^(−x/22.8718)^ − 70.01088 × e^(−x/22.87181)^ (R^2^ = 0.99876). And the release in buffer pH 6.5 and pH 7.4 were also fitted the double exponential diphase kinetics equation of Y_b_ = 111.00884 − 112.97578 × e^(−x/21.51505)^ − 37.91088 × e^(−x/121.77464)^ (R^2^ = 0.99523) and Y_c_ = 158.78318 − 89.86287 × e^(−x/354.59586)^ − 97.88442 × e^(−x/24.58848)^ (R^2^ = 0.99542), respectively. Since the mesoporous/macroporous hollow microspheres had good adsorption of atazanavir in their open pore structure, and the hydrophilic character of sericin guarantied the interactions of hydrogen-bond interactions, π-π stacking and van der Waals force, forming during the atazanavir adsorbed inside the pores, which held back the release of atazanavir. Therefore, this formulation containing meso-/macropores exhibited a slower release.

## 4. Conclusions

Sericin-dextran conjugates and their microparticles were prepared in this study. It was testified that the organic solvents and SDC concentration had a great influence on the size and the morphology of the particles. Microparticles with different shape and size of microspheres, micro-sticks and micro-blocks could be adjusted by different organic solvents. Heterogeneous microspheres could be prepared in ethanol with the range of 85–390 nm, and hollow microspheres in n-propanol. The solubility of atazanavir was improved to 2.22 mg/mL in buffer solutions at pH 2.0 and 1.65 mg/mL in buffer solutions at pH 7.4 by SDC microspheres. The in vitro release studies of atazanavir were carried out based on hollow microsphere with meso-macroporosity for good adsorption and storage. The in vitro release of atazanavir from hollow microspheres of SDC showed the lowest linear cumulative release in acid buffer (pH 8.0), and the most rapid double exponential diphase kinetic cumulative release in buffer (pH 2.5), with the improved aqueous solubility of atazanavir. By this study, we hoped to find a new drug carrier from sericin for improving the aqueous solubility of insoluble drugs.

## Data Availability

Some data are available from the corresponding author by request.

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
