# Peer review of "Microparticles of Sericin-Dextran Conjugate for Improving the Solubility of Antiviral Drug"

_jfb, 2023, doi:10.3390/jfb14060292_

Round 1
Reviewer 1 Report
Comments:
This manuscript (jfb-2401870) by Chen et al., described a nanoparticle comprised of sericin-dextran conjugates for the load of atazanavir, one of antiviral hydrophobic agent. Overall, I ask major revision to authors to be accepted in Journal of Functional Biomaterials due to the below reasons.
1. In Figure 1, just appearance of new band in the FT-IR cannot guarantee that it is chemically conjugated. Please explain the FT-IR data in more detail to confirm whether it is really chemically conjugated.
2. In Figure 2, please assign all peaks in more detail.
3. In Figure 3, please explain the data in more scientifically
4. GPC data is described without showing the graph. Please show the graph as a figure. According to the Mn and Mw data, only one sericin is conjugated to one dextran?
There are so my typo
Author Response
Reviewer 1:
This manuscript (jfb-2401870) by Chen et al., described a nanoparticle comprised of sericin-dextran conjugates for the load of atazanavir, one of antiviral hydrophobic agent. Overall, I ask major revision to authors to be accepted in Journal of Functional Biomaterials due to the below reasons.
- In Figure 1, just appearance of new band in the FT-IR cannot guarantee that it is chemically conjugated. Please explain the FT-IR data in more detail to confirm whether it is really chemically conjugated.
A1: Thank you for your comment. The FT-IR data was discussed in more detail. All revisions have been added to the revised manuscript, marked with red.
Fig. 1a showed that dextran had six characteristic peaks at 3420, 2925, 1647, 1047 and 546 cm-1. Among them, 3420 cm-1 was attributed to the stretching vibration of a large amount of hydroxyl group (-OH), 2925 cm-1 to the -CH2- symmetric stretching vibration, 1647 cm-1 to the C-O stretching vibration, 1047 cm-1 to the α-(1→6) glycosidic bond, and 546 cm-1 to the pyranose ring skeleton. Fig. 1b showed that the characteristic peaks of sericin are 3320 cm-1 and 3067 cm-1 (N-H), 1659 cm-1 (amide I), 1533 cm-1 (amide II), 1251 cm-1 (amide III) and 648 cm-1 (amide V). From the spectrum of sericin/glucan mixture (Fig. 1c), it could be seen that the stretching vibration peak of a large number of hydroxyl (-OH) groups in dextran, the absorption peak of α-(1→6) glucoside bond at 1047 cm-1, and the stretching vibration peak of pyranose ring skeleton at 546 cm-1. There were characteristic peaks of 1659 cm-1 (amide I), 1533 cm-1 (amide II) and 1251 cm-1 (amide III) in sericin. The above results were similar to the native polymer spectra of dextran and sericin. The sericin-dextran conjugate showed absorptoin bands at 2929, 1659, 1535, 1402, 1263, 1151, 1014, 761 and 552 cm-1 (Fig.1d), new bands appeared at 1014 cm−1 and 1151 cm−1 compared with sericin, belonging to the C-O-C stretching vibration in the -COO, confirmed the successful synthesis of the sericin-dextran conjugate.
- In Figure 2, please assign all peaks in more detail.
A2: Thank you for your comment. The 1H NMR spectrum was discussed in more detail. All revisions have been added to the revised manuscript, marked with red.
Fig. 2 was the 1H NMR spectra of dextran, sericin, and sericin-dextran conjugate. Fig. 2a showed the 1H NMR spectrum of dextran, the proton peaks at 4.99 ppm belonged to the 1-H proton characteristic peak, 4.00 ppm belonged to the 6-H, 3.91 ppm belonged to the 5-H, 3.70 - 3.77 ppm belonged to 3, 4-H proton peak, and 3.52-3.59 ppm belonged to 2-H proton characteristic peak. Fig. 2b showed the 1H NMR spectrum of sericin, the proton peaks at 6.82, 6.88, 7.19 and 7.28 ppm belonged to the aromatic ring signal in the tyrosine residue, 8.44 and 8.55ppm belonged to the proton characteristic peak of - (CONH) in the tyrosine and serine in sericin. The proton peaks at 3.89, 4.01 and 4.52 ppm belonged to the characteristic peak of -CH in tyrosine and serine. The proton peaks at 2.07, 2.78 and 2.99 ppm belonged to the characteristic peak of -CH2 in serine and aspartic acid. Fig. 2c showed the 1H NMR spectrum of the sericin-dextran conjugate. The proton peaks belonging to the aromatic ring of tyrosine residues in sericin shifted from 6.82, 6.88, 7.19, 7.28 ppm to 6.83, 7.12, 7.29, 7.47 ppm. Moreover, the characteristic peak of -(CONH) in tyrosine residues also decreased from 8.44 ppm to 8.61 ppm, and from 8.55 ppm to 8.69 ppm. The shift of these proton peaks might be the result of the shielding effect of the triazine ring on the tyrosine residue. The 1-H proton peak at 4.99 ppm and the 2, 3, 4-H proton peak at 3.51-3.76 ppm of dextran were also observed in the spectrum. These results indicated the successful synthesis of sericin-dextran conjugate.
- In Figure 3, please explain the data in more scientifically
A3: Thanks for your suggestion. The detailed captions were added for all the figures. The modification has been marked in red in revised manuscript.
- GPC data is described without showing the graph. Please show the graph as a figure. According to the Mn and Mw data, only one sericin is conjugated to one dextran?
A4: Thank you for your comment. The graphs of GPC were available in Supporting Information (Figure S1, Figure S2, Figure S3). The graft ratio of sericin conjugated to dextran was close to one, according to the number-average molecular weight (Mn) and weight-average molecular weight (Mw) of sericin-dextran conjugate was 9814 and 13833.

Reviewer 2 Report
In this study, Chen et al. developed sericin-dextran conjugates (nano/microparticles) to enhance the solubility of an antiviral drug atazanavir. I have few concerns regarding this study.
1) The authors has to specify the nano/micro system in title.
2) Throughout the manuscript, especially abstract and conclusion, the use of nano and micro terms are very much confusing. Kindly sort it out. For example: In section 3.5, "The lowest cumulative release of atazanavir from the nanoparticles was investigated in basic buffer....". In some places the authors have written microspheres and in some places the authors called microspheres as nanoparticles. This is a major concern, revise the paper thoroughly.
3) "Figure 8. In vitro release of atazanavir from hollow microsphere of SDC in different buffer solutions, 330 (a) pH 2.5, (b) pH 6.5, (c) pH 7.4, (d) pH 8.0." where is drug release study for nanoparticles?
4) Mention about particle size of microsphere in abstract.
5) Why solubility studies for atazanavir pure drug has not been performed in different pH conditions?
6) Why dissolution studies have not been conducted for atazanavir pure drug as a comparison to SDC nanoparticles and microspheres?
7) Reduce the conclusion part, make it concise and informative.
Need substantial improvement in English language along with proper formatting as I could see difference in font size of multiple paragraphs.
Author Response
Reviewer 2:
In this study, Chen et al. developed sericin-dextran conjugates (nano/microparticles) to enhance the solubility of an antiviral drug atazanavir. I have few concerns regarding this study.
1) The authors has to specify the nano/micro system in title.
A1: Thanks for your suggestion. The title has been modified in revised manuscript.
2) Throughout the manuscript, especially abstract and conclusion, the use of nano and micro terms are very much confusing. Kindly sort it out. For example: In section 3.5, "The lowest cumulative release of atazanavir from the nanoparticles was investigated in basic buffer....". In some places the authors have written microspheres and in some places the authors called microspheres as nanoparticles. This is a major concern, revise the paper thoroughly.
A2: Thanks for your comment. I am so sorry for the confusion of nano and micro terms. All the descriptions were unified as microparticles in the revised manuscript.
3) "Figure 8. In vitro release of atazanavir from hollow microsphere of SDC in different buffer solutions, 330 (a) pH 2.5, (b) pH 6.5, (c) pH 7.4, (d) pH 8.0." where is drug release study for nanoparticles?
A3: Thanks for your comment. Because of the better loading rate of hollow microsphere, hollow microsphere of SDC was chosen as the carrier to study the long-term release of atazanavir from this carrier.
4) Mention about particle size of microsphere in abstract.
A4: It has been revised in the article. Heterogeneous microspheres could be prepared in ethanol with the range of 85-390 nm, and hollow mesoporous microspheres in propanol with an average particle size of 2.5-22 µm.
5) Why solubility studies for atazanavir pure drug has not been performed in different pH conditions?
A5: Dissolution experiments of pure atazanavir and atazanavir-loaded SDC microparticles were carried out in buffer solutions at pH 2.0, 6.5, 7.4, and 8.0, as showed in Fig. 7. The release of atazanavir from the native drug tablets had a highly pH dependent aqueous solubility. It showed high solubility of 2. 21 mg/mL at pH 1.0, and it decreased rapidly to 0.29 mg/mL with increasing pH at pH 2.0, and 0.041 mg/mL at pH 3, and it showed nearly no release with increasing pH (Fig. 7A).
6) Why dissolution studies have not been conducted for atazanavir pure drug as a comparison to SDC nanoparticles and microspheres?
A6: Thanks for your comment. The release of atazanavir from the native drug tablets had a highly pH dependent aqueous solubility. There was nearly no solubility of atazanavir in PBS at pH 6.5, pH 7.4, pH 8.0. Therefor the dissolution experiment of pure atazanavir was not available.
7) Reduce the conclusion part, make it concise and informative.
A7: Thanks for your comment. The conclusion part was revised in the manuscript.
Need substantial improvement in English language along with proper formatting as I could see difference in font size of multiple paragraphs.
A8: Thanks for your comment. The manuscript has been corrected for the format and language.

Reviewer 3 Report
This is an interesting study about sericin-dextran conjugate for improving the solubility of antiviral drug. However, it could be considered to be published only after the following points are well addressed.
1. Line 162-170, it is surprising that the Mn of dextran and sericin is only around 5200 and 3400. The authors should confirm this result.
2. The zeta potential of the dextran-sericin conjugate loading with and without drugs should be measured.
3. In the introduction (before line 55-59), polymer-protein conjugate should be introduced. Several papers related to this point (Journal of Materials Chemistry B 1 (19), 2482-2488; Biomacromolecules 23.11 (2022): 4948-4956) should be included.
4. The quality of figure 4 and 7 could be improved to a higher level.
5. Technical issue. For example, line 21-29, the font size of the text is not uniform. Please check all.
Moderate editing of English language
Author Response
Reviewer 3:
This is an interesting study about sericin-dextran conjugate for improving the solubility of antiviral drug. However, it could be considered to be published only after the following points are well addressed.
1. Line 162-170, it is surprising that the Mn of dextran and sericin is only around 5200 and 3400. The authors should confirm this result.
A1: The dextran used in this study was obtained from Shanghai Huamao Pharmaceutical Co., Ltd, with the name of Dextran-5 (Mn 4000-6000). Sick sericin was purchased from Huzhou Attest Biochemical Co. Ltd, with the Mn of 2000-3000.
2.The zeta potential of the dextran-sericin conjugate loading with and without drugs should be measured.
A2: Thanks for your suggestion. The zeta potential of the microparticles of dextran-sericin conjugate loading with and without drugs were measured and added in revised manuscript.
3.In the introduction (before line 55-59), polymer-protein conjugate should be introduced. Several papers related to this point (Journal of Materials Chemistry B 1 (19), 2482-2488; Biomacromolecules 23.11 (2022): 4948-4956) should be included.
A3: Thanks for your suggestion. The related papers had been included in revised manuscript.
4.The quality of figure 4 and 7 could be improved to a higher level.
A4: Thanks for your suggestion. Figure 4 and Figure 7 had been improved in revised manuscript.
5. Technical issue. For example, line 21-29, the font size of the text is not uniform. Please check all.
A5: Thanks for your comment. The manuscript has been corrected for the format and language.

Round 2
Reviewer 1 Report
The authors have fully and sincerely responded to the reviewer's questions and requisition. I agree that this manuscript is ready to be published in the JFB.
Reviewer 2 Report
The authors have addressed all my queries. This manuscript can be accepted for publication after English language editing.
Minor English language editing is required.
Reviewer 3 Report
Accept in present form
Minor editing of English language required